# Effect of the Addition of Humic Substances as Growth Promoter in Broiler Chickens Under Two Feeding Regimens

**DOI:** 10.3390/ani9121101

**Published:** 2019-12-09

**Authors:** Alejandra Domínguez-Negrete, Sergio Gómez-Rosales, María de Lourdes Angeles, Luis Humberto López-Hernández, Tercia Cesaria Reis-de Souza, Yair López-García, Anai Zavala-Franco, Guillermo Téllez-Isaias

**Affiliations:** 1Faculty of Natural Sciences, Autonomous University of Queretaro, Av. de las Ciencias S/N, Juriquilla, Queretaro 76230, Mexico; mvzaledom@gmail.com (A.D.-N.); tercia@uaq.mx (T.C.R.-d.S.); 2National Center of disciplinary Research in Animal Physiology and Genetics, INIFAP, Km 1 carretera a Colon Ajuchitlán, Queretaro 76280, Mexico; angeles.lourdes@inifap.gob.mx (M.d.L.A.); lopez.lhumberto@inifap.gob.mx (L.H.L.-H.) Yair.lopez.garcia@outlook.com (Y.L.-G.); 3Center for Research and Advanced Studies of the National Polytechnic Institute, CINVESTAV-IPN, Real de Juriquilla, Queretaro 76230, Mexico; anai.zavala@cinvestav.mx; 4Department of Poultry Science, University of Arkansas, Fayetteville, AR 72701, USA; gtellez@uark.edu

**Keywords:** broilers, humic substances, productivity, lactobacillus, clostridium, coccidian oocysts

## Abstract

**Simple Summary:**

The rapid spread of antimicrobial-resistant genes in bacterial communities is a threat to human, animal, and environmental health that continues to progress inflexibly. Humic substances (HS) are promising complex molecules as an alternative to reduce the use of growth promoter antibiotics (GPA) in animal feeds. Improvements in productivity, intestinal health, immune response, and antioxidant status have been reported in broilers supplemented with HS. In the present study, broilers fed with an extract of HS (EHS) from a worm compost had similar carcass yield and excretion of coccidian oocysts but increased *Clostridium perfringens* and lactic acid bacteria (LAB) compared to broilers fed diets added with GPA. Broilers subjected to feed restriction had reduced growth performance and meat quality. These results confirm the growth-promoting effect of ESH, which could be explained by higher abundance of the beneficial LAB and by reducing the multiplication of harmful parasites in the gut of broilers.

**Abstract:**

Humic substances (HS) from different sources have been evaluated to replace or reduce the use of growth promoter antibiotics (GPA) in the feeds of broiler chickens. The objective was to evaluate the growth performance, tibia measurements, nutrient balance, meat quality, and microbiological status of broiler fed with an HS extract (EHS) under ad libitum (ADLIB) or feed restriction (REST). Individually caged broilers (*n* = 180, 14–35 day of age) were assigned to a factorial arrangement of three dietary treatments: (1) positive control with bacitracin methylene disalicylate (BMD) and salinomycin; (2) negative control without BMD nor salinomycin, and (3) same as negative control with 0.25% EHS, and two feeding regimens 1) ADLIB or REST for 24 h on d 1, 7, and 14. Results were subjected to ANOVA. Positive control and EHS-fed broilers showed higher carcass yield (*p* < 0.05) and lower oocyst excretion (*p* < 0.01) compared to negative control birds. Lactic acid bacteria (LAB) and *Clostridium perfringens* (*C. perfringens*) were higher in negative control and EHS-broilers compared to positive control (*p* < 0.01). In conclusion, higher carcass yield, lower *C. perfringens* and oocyst excretion were found in positive control and higher carcass yield, higher LAB and lower oocyst excretion were found in EHS-fed broilers. Broilers subjected to REST had reduced growth performance and meat quality. In conclusion, EHS could be used to increase the carcass yield and beneficial LAB in broilers.

## 1. Introduction

Antimicrobial resistance threatens the very core of modern medicine and the sustainability of an active, global public health response to the enduring threat from infectious diseases, and has triggered the development of coordinated and comprehensive national and global action plans [1,2]. Antimicrobial resistance can pass between different bacterial populations through humans, livestock, and the full environment [3]. Farm animals are an essential component of this triad because they are exposed to growth promoter antibiotics (GPA) to improve animal health, welfare, and productivity, and act as reservoirs and vectors of resistance genes [1,3]. For this reason, several new additives have been tested as growth promoters to reduce or alternate the inclusion of GPA in feeds, while maintaining an efficient animal production to obtain safe edible products [4,5,6].

In veterinary practice, humic substances (HS) have been used as antidiarrheal, analgesic, immunostimulatory, and antimicrobial agents in Europe [7,8]. The main substances that make up the HS are the humic acids (HA), fulvic acids (FA), and humins, and hence, they are a complex mixture of aliphatic chains or aromatic rings with specific content of functional groups, but the concentration of these substances differs according to the raw materials that they originate from [8,9,10]. Leonardite and lignites, two non-renewable resources, are the primary commercial sources of HS, while compost and worm compost, especially those produced with animal manure, are two environmentally friendly sources of HS [4,5,6].

During the last two decades, different sources of HA, FA, and HS have been tested as growth promoters in the feeds or the drinking water of poultry [4,8]. In broiler chickens, improvements in body weight, feed conversion, the ash content of the tibia, and the retention of ashes, nitrogen, and energy as well as reduced crypt depth and increased length of the villi of the jejunal mucosa due to the inclusion of HS have been reported [4,8]. Increased antioxidant activity in birds reared under normal and stress conditions [11,12] and lower *Escherichia coli* counts in the digesta of the small intestine have also been reported in broilers added with HS (50). However, contrasting results have also been found. An important issue to consider about the supplementation of HS is the differences in the content of HA and FA, the inclusion level and form (water or feed) and characteristics such as chain length, side-chain composition and origin (plant, soil, peat, and coal-derived) of HS previously used in poultry [4,8,10].

In previous studies, HS derived from a worm compost was supplemented to broilers through the drinking water or feeds with encouraging results [4,5]. Broilers added with a worm compost leachate in the drinking water showed lower feed conversion ratio (FCR) and higher energy digestibility and higher retention of dried matter, ashes, nitrogen, and energy [4]. Chicks fed with HA extracted from the same worm compost as the one used in this study, and subjected to feed restriction (REST) for 24 h showed higher intestinal viscosity and lowered bacterial liver translocation and serum concentrations of fluorescein isothiocyanate dextran (FITC-d), a marker of gut permeability [13,14], compared to control chicks. These findings suggest that HS from a worm compost may promote the growth of poultry by increasing the digestibility and retention of dietary components and by reducing the passage of bacteria from the intestine into the body. The objective of this research was to evaluate the growth performance, tibia measurements, nutrient balance, meat quality, and microbiological status of broiler chickens fed an extract of HS (EHS) and subjected to a program of feed restriction to induce intestinal inflammation.

## 2. Materials and Methods

This research was revised and approved by the Bioethics Committee of the Faculty of Natural Sciences of the Autonomous University of Queretaro (Number 108FCN2018).

### 2.1. Characterization of the EHS

The extraction of HS from the worm compost was performed as previously described [8] following an alkaline extraction process. Sodium hydroxide (NaOH 0.5 M) and the worm compost were mixed in a ratio of 5:1 (mL/g) in 50 mL tubes and allowed to stand for 24 h at room temperature. After this, the tubes were centrifuged for 20 min at 3000× *g* (5810R Eppendorf centrifuge, Hamburg, Germany), and the precipitate and supernatant were separated by decantation. The precipitate was washed twice in distilled water and centrifuged as before. The supernatants were pooled, dried in a forced air stove (Shel Lab, Cornelius, OR, USA) at 55 °C for 24 h, and ground using a Thomas Willey grinder and 1 mm sieve. For the identification of functional groups, a Perkin-Elmer Spectrum GX infrared spectrophotometer with Fourier transformation with attenuated total reflectance (FTIR-ATR) with an ATR accessory was used. All spectra were collected in a range of 4000–400 cm^−1^, with a resolution of 4 cm^−1^, using 32 scans per sample. The sample was analyzed in triplicate. Elemental analysis of EHS was carried out using energy dispersive X-Ray spectroscopy (EDS). An environmental scanning electron microscope Phillips XL30/40 EDS-ESEM equipped with an X-ray source (XTrace microspot) and an XFlash^®^6/10 detector (Bruker Nano GmbH) was used. The sample was analyzed in triplicate, using zone mapping. Crystal types were detected with X-ray diffraction (XRD). Samples were analyzed in triplicate on a 2100-Rigaku diffractometer. A 30 kV CuKα radiation source and a fixed 20 mA power source were used. The diffraction patterns were measured in a 2-Theta region from 3 to 70°, using a 0.02° step. Results were used for the calculation of aromaticity. The aromaticity was figured out based on the method described before [15]. The γ and G bands were identified, representing the aliphatic and aromatic parts, respectively. Deconvolution of the XRD pattern obtained by the PeakFit v.4.12 program was performed, and the area under the curve was determined using the following formula:$$\mathrm{Aromaticity},\;\%\;=\;\frac{\mathrm{Band}\;G\;\times 100}{\mathrm{Band}\;G+\mathrm{Band}\;\gamma }\;.$$

### 2.2. Animal, Treatments, and Diets

A group of 1-d old Ross 308 male broilers were obtained from a regional commercial hatchery and randomly allocated in floor pens with wood shaving and gas brooder heaters. Chickens were vaccinated *in ovo* against Newcastle disease, Marek’s disease, and Laringotracheitis. The first week the temperature was kept at 32 °C, and then, it was reduced 2 °C each week until the third week; from the fourth week on, the temperature was maintained between 21 and 23 °C. During the first seven days a photoperiod of 23 h of light and 1 h of darkness was maintained, and after eight days, the cycle was maintained at 20 h light and 4 h of darkness.

From 1 to 13 d after hatching, chicks were fed a standard broiler starter diet in mash form (Table 1). On day 10, 180 birds were penned individually in holding mesh-floored cages with a metal feeder and a cup drinker in a manually ventilated unit to adapt them to the facilities before starting the experiment. At day 14, broilers were randomly assigned to a factorial arrangement of three dietary treatments: (1) positive control with bacitracin methylene disalicylate (BMD) and salinomycin; (2) negative control without BMD nor salinomycin, and (3) same as negative control without BMD nor salinomycin and added with 0.25% EHS, and two feeding regimens (1) ADLIB or REST for 24 h on d 1, 7, and 14 of the study. The experiment lasted 21 d and during this time broilers were kept in the cages, from 14 to 35 d of age. The diets were formulated with corn and soybean meal and were fed in mash form (Table 1).

### 2.3. Sample Collection and Laboratory Determinations

Birds were weighed at the beginning and end of the trial to calculate the daily weight gain (WG, g/d). Feed offered and refused was registered to calculate the daily feed intake (FI, g/d). The feed conversion ratio (FCR) was estimated by dividing the FI between the WG. During the last three days of the experiment, total excreta were collected from 15 broilers per treatment. During this period, the feed offered was divided into two equal daily meals to minimize feed spoilage. The cages were equipped with side metal walls and underneath trays, in which a plastic bag was placed to collect the excreta every 24 h. The excreta were weighed and stored frozen at −20 °C. At the end of the experiment, droppings were collected from each cage and pooled to get one replicate sample out of three broilers from the same treatment for oocyst counting. Then, all broilers were slaughtered by cervical dislocation to collect the ileal and cecal content in sterile plastic bags for microbial analysis. The ileal and cecal content from three broilers were pooled to get one replicate sample per treatment. The breast, legs, thighs, and carcass yield were also estimated. The left tibia was removed and frozen.

Excreta samples were lyophilized and ground using a 2 mm mesh. Determinations of dry matter, ashes, nitrogen, and energy were carried out in the diets and excreta to estimate the nutrient balance. All laboratory determinations were carried out following standard procedures, according to the Association of Official Analytical Chemists (AOAC) [16]. The apparent metabolizable energy corrected to zero nitrogen retention (AMEn) was also estimated. Tibias were unfrozen, cleared of soft tissues, weighed, dried using a horizontal flow drying oven (Terlab S.A. de C.V., Zapopan, Jalisco, México) at 105 °C for 24 h, defatted in ethyl ether, and incinerated in a furnace (Furnatrol I Type 1,8200; Thermolyne, Guadalajara, Jalisco, Mexico) at 600 °C for 6 h to determine the contents of ash [17].

After broilers were killed and exsanguinated, the breast was excised and placed in a chilling room until reaching a temperature of 4 °C. The pH was measured using a punction glass electrode connected to a HI 99163 meat pH meter (HANNA Instruments Mexico., Mexico City, Mexico.). The exterior and internal color of the breast meat was measured 30 min after bleeding using a Minolta reflectance colorimeter and reported in the CIE system values of lightness (L*), redness (a*), and yellowness (b*). The water holding capacity was assessed using three methodologies. First, the filter paper method with some modification was used [18]. In brief, a sample of 0.3 g of meat was placed in a weighed Whatman filter paper (No. 541, 110 mm diameter) and pressed with a 2.5 kg plate for 5 min. After removing the piece of meat, the filter paper was reweighed and the percentage of weight loss before and after pressing was calculated. For the centrifugation method, 5 g of ground meat was weighed in a tube, 8 mL of NaCl were added with stirring for 1 min; then, the tubes were held in cold water bath for 30 min and centrifuged (5810R Eppendorf centrifuge, Hamburg, Germany) at 12,000× *g* for 15 min. The supernatant was measured in a 10 ml tube, and by difference, the retained NaCl in the meat was quantified as a percentage [19]. The drip loss was determined in a piece of breast meat of approximately 100× *g* [20]. Samples were placed in a sealed polyethylene bag, stored for 24 h at chill temperature (4 °C), and reweighed. Drip loss was reported as a percentage of the weight difference of the sample before and after chilling.

Two techniques were used to evaluate antioxidant activity. The determinations of 1,1-diphenyl-2-picrylhydrazyl (DPPH) radical scavenging activity and ferric radical antioxidant power (FRAP) were carried out in extracts of 5× *g* of meat. In brief, the sample was homogenized in 25 mL of phosphate buffer (IKA homogenizer T25) for 1 min. The homogenate was centrifuged for 30 min at 12,000× *g* at 4 °C and then filtered on Whatman No. 4 filter paper. The filtrate was collected in Eppendorf tubes and stored at −20 °C for further analysis. For DPPH determinations [21], 25 μL of the extract was mixed with 975 μL of the DPPH solution in test tubes and incubated in the dark for 1 h at room temperature. The absorbance was read at 515 nm. The results were expressed as mmol equivalent of Trolox/kg of meat. For FRAP determinations [21] an aliquot of 25 μL of the extract was mixed with 975 μL of a FRAP solution (2.5 mL of 2,4,6-tripyridyl-s-triazine acid (TPTZ)) 40 mM, 2.5 mL of FeCl_3_ 20 mM, and 25 ml of acetate buffer 0.3 mM, pH 3.6) in test tubes. The absorbance was read at 593 nm using a UV/VIS spectrophotometer (GENESYS 10S UV-Vis Thermo scientific), and the readings were taken at 0 and 6 min from the start of the reaction. Results were expressed as mmol equivalent of Trolox/kg of meat.

The lipid oxidation was measured using the thiobarbituric acid-reactive substances (TBARS) test [22]. For this, 5 g of meat were homogenized in an ice bath with 20 mL of trichloroacetic acid (TCA) solution for 1 min. The homogenate was centrifuged for 20 min at 12,000× *g* at 4 °C. The supernatant was filtered in light protected tubes and stored at −20 °C for further analysis. TBARS were determined in 1 mL of the extract mixed with 1 mL of TBA in test tubes. Tubes were heated for 30 min at 95 °C cooled, centrifuged, and the absorbance was read at 530 nm. Results were expressed as mg of malondialdehyde (MDA) kg of meat.

Ileal samples were used for lactic acid bacteria (LAB) counting in dilutions of 1:1–7 wt/vol with 0.01% peptone water [23]. Aliquots of 100 µL were added to Petri dishes with Man Rogosa Sharpe medium (MRS) (DIBICO S.A. de C.V., Mexico City, Mexico). Plates were incubated (Model Max Q4450, Thermo Scientific, Mexico City, Mexico) at 35 °C for 48 h in a microaerophilic atmosphere (5% O2) using a microaerophilic container system (GasPak EZ, BD Diagnostics, Sparks MD, USA). Results were expressed as log_(10)_ of colony-forming units per gram (log_(10)_CFU/g). Cecal samples were diluted in 1:1–7 wt/vol with 0.01% peptone water for *Clostridium perfringens* (*C. perfringens*) counting [24]. Aliquots of 100 µL were added to Petri dishes with tryptose-sulfite-cycloserine (TSC) culture medium (DIBICO S.A. de C.V., Mexico D.F.) and egg yolk emulsion (Thermo Fisher, Scientific Inc., Mexico City, Mexico). Plates were incubated for 48 hours in gas-pack anaerobic jars at 37 °C. Colonies were identified and counted (log_(10)_CFU/g). The number of oocysts in excreta was assessed using the McMaster counting chamber technique [25].

### 2.4. Statistical Analysis

Data were subjected to an analysis of variance, following the procedures of the general linear models [26], using a 3 × 2 factorial design with dietary treatments and feeding regimen as main effects plus their interaction. Per treatment, 30 replicates for the growth performance, carcass, and tibia measurements, 15 replicates for the nutrient balance, and meat quality and ten replicates for microbial determinations, and oocyst counts were used. Percentage results were transformed to arcsine values before analysis. The results were reported as least square means and standard error of the mean (SEM).

## 3. Results

### 3.1. Composition of Humic Substances

The concentration of HA, FA, and ashes in the EHS were 47.1%, 29.6%, and 23.2%, respectively, on dry matter basis. The estimated aromaticity of EHS was 53.8%. The infrared spectra of the EHS are depicted in Figure 1, and the relative intensity of the main bands and associated functional groups of the EHS are presented in a Appendix A. The elemental composition of EHS is given in Table 2.

### 3.2. Effect of Dietary Treatments

There were not any differences due to the interaction of the dietary treatment and feeding regimen in any of the variable responses evaluated in the present study (*p* > 0.05). In regards to dietary treatments, the growth performance and yield of breast, thighs, and legs did not show any differences among the positive control, negative control, and EHS-fed broilers (Table 3). The carcass yield was lower in broilers fed the negative control diet (40.3%; *p* < 0.05; SEM: 0.177) compared to those fed the positive control (40.9%) and EHS diets (40.9%); between the positive control and EHS-fed broilers no differences were observed on the carcass yield. Fresh weight, dry weight, and ashes weight of the tibia were similar (*p* > 0.05) among treatments (Table 4). The intake, excretion, and retention of dry matter, ashes, nitrogen, and energy, as well as the AMEn were similar among broilers of the different treatments (Table 4).

Similar pH, exterior and internal breast meat color, water holding capacity using the filtration, centrifugation, and dripping method, as well as the index of lipid oxidation (TBARS) and antioxidant status (DDPH and FRAP) of breast meat, were found among treatments (Table 5). The LAB counts in the ileal digesta were lower in positive control broilers (5.68 log(10) UFC/g; *p* < 0.01; SEM: 0.200) compared to the negative control (6.46 log(10) UFC/g) and EHS-fed broilers (6.99 log(10) UFC/g); and EHS-fed broilers showed higher LAB compared to negative control broilers (Table 5). The *C. perfringens* counts in the ceca were lower (*p* < 0.01) in the positive control (6.55 log(10) UFC/g; *p* < 0.01; SEM: 0.161) compared to the negative control (7.25 log(10) UFC/g) and EHS-fed broilers (7.15 log(10) UFC/g); and between the negative control and EHS-fed broilers no differences in C. perfringens count were observed (Table 5). The oocyst number in excreta was higher in negative control broilers (194.4 oocysts/g; *p* < 0.01; SEM: 24.197) compared to positive control (11.7 oocysts/g) and EHS-fed broilers (13.9 oocysts/g); between the positive control and EHS-fed broilers no differences on the oocyst excretion were observed (Table 5).

### 3.3. Effect of Feeding Regimen

The feeding regimen had several significant effects on the growth of broilers. The final body weight, FI, and WG, as well as the legs and thighs weight and the breast and carcass weight and yield were lower (*p* < 0.01) in REST broilers compared to ADLIB broilers (Table 3). The fresh weight, percentage dry matter, dry weight, and ashes weight of the tibia were lower (*p* < 0.05) in REST compared to ADLIB broilers (Table 4). Additionally, the excretion of dry matter (*p* < 0.05) and nitrogen (*p* < 0.01) were higher and the retention of nitrogen (*p* < 0.05) was lower in REST broilers; other variables were not different between ADLIB and REST broilers (Table 5).

## 4. Discussion

### 4.1. Composition of Humic Substances

The estimated aromaticity of EHS was 2.5 times higher than that reported for feces of common earthworm species responsible for producing humus in soil [27]. The broad absorption regions of infrared spectroscopy showed by EHS fluctuated from 3467 to 774 cm^−1^ (Figure 1) The infrared spectra and functional groups of the EHS are inside the range reported in previous studies in which HS from soils, cow dung, worm compost, sludge, sediments, and lignite coals were characterized [28,29,30]. Group composition is used to characterize HS as it gives information about their chemistry and structural properties [31], and for which they are considered the most widely-spread natural complexing ligands occurring in nature, with high capacity to form aggregates within solutions and outstanding ability to participate in redox reactions [9,32]. The acidity and degree of polymerization of HS changed systematically with increasing molecular weight. The main elements of EHS (Table 2) were O (45.8%), Na (27.5%), and C (13.8%). Small amounts of Si (4.4%), K (3.88%), Cl (2.9%), and S (1.5%) were observed. Very low concentrations of P (0.81%) and Ca (0.43%) were also found. The concentration of O was inside the range, C was below the range, but Na was above the values reported previously for different sources of HS such as different types of soils, cow dung, worm compost, sludge, sediments, and lignite [28,33,34]. The unusually high content of Na might be explained by the use of NaOH for the extraction of HS. Unusual high K content has also been observed in HS extracted with KOH [34]. The content of Si, K, Cl, S, P, and Ca also depends on the organic materials that EHS originate from [8,9,10]. These elements are normal components of animal feeds, and hence, of animal manure.

### 4.2. Effect of Dietary Treatments

Several mechanisms of action have been proposed to explain the benefits observed in broiler chickens supplemented with HS, including the ability to create protective layers over the epithelial mucosal membrane of the digestive tract against the penetration of toxic and other bacterial contaminated substances [4,5,8]; whilst, feed restriction for 24 h has been successfully used to induce intestinal inflammation as a means to disrupt the tight junctions of enterocytes [13,14].

In previous research [5] chicks fed with HA extracted from the same worm compost as the one used in this study and subjected to REST for 24 h showed higher intestinal viscosity and lowered bacterial liver translocation and serum FTIC-d compared to control chicks. Higher intestinal viscosity, and hence, the ability of HS to create protective layers, have been linked to the high capacity of HS to form aggregates within solutions [10]. In the present experiment, the previous findings were expected to be confirmed under the assumption that REST broilers added with ESH may have had better productive performance compared to REST positive and negative control broilers. However, due to the lack of interactions of the dietary treatment and feeding regimen in any of the variable responses evaluated in the present study our hypothesis cannot be validated.

The lack of differences in the final body weight but higher carcass yield in EHS-fed broilers in the present study agreed with a recent study in which a lack of effect on the final body weight, but higher breast weight and yield were observed in broilers added with potassium humate [35]. Our results indicate that the FI and the intake of nutrients (see Table 5 below), the WG, and FCR were similar among treatments, but it might be that these nutrients were used differently inside the body. It seems that in the positive control and EHS-fed broilers, the amount of nutrients used for the synthesis and storage of meat was similar, but this amount was lower in the negative control broilers, which led to lower carcass yield. It is probably that in negative control broilers higher amounts of nutrients were diverted to strengthen the immune response due to the removing of GPA from the feed that may increase the risk of bacterial challenges in the gut [3,36]. Nevertheless, the authors were aware that there was no a clear explanation of these results and the topic requires further clarification.

The colloidal characteristics of HS and their ability to form chelates with different ions have been related to improved mineral use in plants and animals [10]. In previous studies, higher tibia ash [37] and higher ash retention [4] were reported in broiler chickens supplemented with HS. In laying hens and pheasants supplemented with HS improved percentage, thickness, and hardness of eggshell were also reported [38]. These findings did not agree with the lack of effects of the EHS on the tibia measurements in our research (Table 5). Regarding the balanced dietary components, in broilers supplemented with worm leachate as a source of HS through the drinking water, higher retention of dry matter, ashes, and nitrogen, and higher AMEn were observed [4]. In rats supplemented with HS, higher nitrogen retention has also been observed [39]. Recently, in FA-fed broilers [40] higher activities of the digestive enzymes amylase, lipase, and protease were reported, which agrees with the increased length of the mucosal villi of the jejunum and the increased gut length due to dietary inclusions of HS in broiler [12,41]. However, these findings do not agree with the lack of effects of EHS on nutrient use in our research.

In in vitro studies, the redox properties of HS have been extensively studied. Quinones are considered to be one of the principal reducible moieties, while phenols are rated as the primary electron-donating moieties with antioxidant properties relative to the electron-accepting quinones [32,42,43]. In broiler chickens supplemented with HS, a potent antioxidant activity such as increased glutathione reductase, total antioxidant and catalase activity in the blood has been reported [11]. In broilers supplemented with increasing dietary FA, increased superoxide dismutase and glutathione peroxidase activity and decreased malondialdehyde levels in the blood were also found in a recent study [40]. In broilers supplemented with HS and subjected to transportation stress, increased superoxide dismutase activity in the mitochondria of the liver compared to the control group was also observed [12]. The activities of glutathione reductase followed the same trend. These findings did not agree with the lack of effect of the EHS on the antioxidant status and index of lipid peroxidation in the breast muscle and the meat quality variables of this study. In our research, the antioxidant status of breast meat was measured, rather than in blood, because the meat is the final product for human consumption, and the antioxidant status of the meat may influence the shelf life and other variables such as meat color, flavor, and tenderness [12,44].

Lower ileal and cecal LAB counts have been reported in BMD-added broilers compared to those fed diets without BMD [36,45,46]. In agreement with our results, broilers reared in floor pens and added with HS showed increased LAB counts in the digesta collected from the distal end of the duodenum to the ileocecal junction and cecum compared to those fed a diet added with flavomycin [47]. Previously, in broilers reared in floor pens fed a mined humate compound, a slight increased cecal LAB was also observed [48]. However, in HA-fed broilers, a lack of response to the ceca LAB was reported [5].

In recent reports, no effects on the *C. perfringens* counts in the jejunum, ceca, and colon [48,49] and lower or higher *C. perfringens* counts in the ceca and litter [49,50] have been reported in broilers fed BMD-added diets compared to those fed diets without added BMD. No reports of the effects of HS on *C. perfringens* were found in poultry on published articles. In piglets fed diets added with sodium humates, no effects on fecal *C. perfringens* were observed after 21 d of supplementation [51]. In humans supplemented with a standardized HA preparation, the colonic concentration of *C. difficile* increased, but the observed difference was not significant [52].

This was the first report of a reduction of the oocyst excretion by EHS in poultry. Since broilers were exposed to a natural infestation and were allocated in holding pens, the number of oocyst in excreta shown in Table 5 were far lower than those observed in coccidian vaccinated or challenged broilers [53,54]. This topic deserves further studies to confirm the inhibitory effect of ESH on the excretion of oocysts in coccidian vaccinated or challenged broilers. In the area of plant diseases and pest control, the use of worm compost extracts/teas as antiparasite agents has gained interest during the last two decades [55,56]. In the area of aquaculture, reductions of parasite load and infections have been reported in different types of fishes exposed to HS under pond culture conditions [57,58]. Furthermore, in experimentally challenged mice with *Trypanosoma brucei brucei* and *T. brucei gambiense*, the administration of a humus extract for 21 d in the drinking water induced adequate protection and significantly reduced the mortality rate, while all non-treated control mice died within 10 d after the challenge [59]. The authors suggest that the enhancement of the innate immune system might be involved in host defense against trypanosomiasis. In several in vitro studies, the immunostimulant effects of HS have been demonstrated [60]. In broilers, fed diets containing HS, increased antibody titers against the avian influenza virus [61] and Newcastle virus [47,62] has been reported. It has been suggested that the improved immune response in broilers fed HS-added diets may be due to the greater growth of immune organs such as the thymus and bursa of Fabricius [35]. It may be possible that through improvements of the immune response, the addition of EHS in the present study reduced the oocyst excretion.

A summary of the results indicates that the growth performance, tibia measurements, the balance of dietary components, and the quality and antioxidant status of breast meat were not affected by the dietary addition of the EHS. Discrepancies on these responses compared to previous studies could be to several factors. Some of these factors could be that in our study broilers were allocated in individual cages, the supplementation of the EHS starter at 14 d of age and it was supplemented for only 21 d. In this design, the REST feeding regimen was introduced as a source of stress, focused to cause intestinal inflammation. This design also allowed us to collect the excreta for the balance of dietary components. All these conditions differed from other studies in which broilers were raised in floor-pens and the HS were supplemented from d one of age [4,8,10]. Other important factors to consider are differences in the content of HA and FA, the inclusion level, and form (water or feed) and characteristics such as chain length, side-chain composition, and origin (plant, soil, peat, and coal-derived) of HS previously used in broilers [4,8,10]. It is probably that subtle differences in the composition exist between HS extracted from leonardite and lignites and those extracted from worm composts [4,5,6]. This topic also deserves further studies.

Positive control and EHS-fed broiler chickens showed higher carcass yield compared to negative control birds. In positive control broilers lower *C. perfringens* and oocyst excretion were expected due to the dietary addition of BMD and salinomycin, while in negative control broilers fed without BMD and salinomycin higher *C. perfringens* and oocyst excretion were anticipated. On the other side, EHS-fed broilers had similar *C. perfringens*, higher LAB and lower oocyst excretion related to negative control broilers. Furthermore, positive control broilers showed lower *C. perfringens*, lower LAB and similar oocyst excretion compared to EHS-fed broilers. Although *C. perfringens* is a potentially pathogenic enterobacterium, and higher cecal counts could cause the presence of necrotic enteritis [49], it has been documented that one of the proposed mechanisms to explain the benefits observed in broiler chickens supplemented with HS, is their the ability to create protective layers over the epithelial mucosal membrane of the digestive tract against the penetrations of toxic and other bacterial contaminated substances [4,5]. It is probably that even if *C. perfringens* were higher in EHS-fed broilers the high capacity of HS to form aggregates within solutions, and hence, increased gut viscosity, avoided *C. perfringens* from damaging the intestinal mucosa. It can be suggested that lower carcass yield in negative control fed chickens was due to increased *C. perfringens* counts and higher *Eimeria* oocyst in excreta, which diverted nutrients to the immune response.

### 4.3. Effect of Feeding Regimen

In a previous experiment [5] chicks fed HA were subjected to REST for 24 h and then killed whilst in the present study, REST broilers were subjected to three periods of 24-h of feed restriction, which explains the lower performance, muscle, and skeletal development. Lower pH and higher superficial L and water loss by centrifugation (*p* < 0.05) in breast meat of REST broilers were observed (Table 5). No differences in LAB, *C. perfringens*, and excretion of oocysts were found between REST and ADLIB broilers (Table 5). In ceca tissues from molted hens subjected to 14-d of feed withdrawal, *Salmonella* invasion was increased compared to ceca tissue from controls hens [63] and in chicks subjected to 24-h of feed restriction, increased liver bacterial translocation was observed [5]. Furthermore, increased serum levels of corticosterone and several acute-phase proteins were reported in broilers from 18 to 30 h after the initiation of feed deprivation [64] as indicative of an inflammatory process. In our research, the lack of differences in LAB, *C. perfringens* and excretion of oocysts might have been because samples were taken in broilers in the fed-state. However, the higher mucosal permeability, higher risk of bacterial invasion, stress of hunger, and the transient catabolic state during feed restriction might also explain the deterioration of meat quality and the higher nitrogen excretion in broilers subjected to REST in the present study.

It has been suggested that potential alternative growth promoters should be evaluated under simulated commercial conditions in which broilers are faced with different stress sources. In our study, REST was used to cause intestinal inflammation as a potential risk of increased intestinal permeability to pathogenic bacteria. It was clear that REST caused detrimental effects on the growth and characteristics of the meat, but the addition of ESH did not prevent these negative effects. The beneficial effect of ESH, by reducing the intestinal permeability, was observed before immediately after a 24-hour restriction [5], but in our study we tried to mimic an intermittent chronic-type source of stress by applying REST during three times (at the beginning of each week). The results suggest that the addition of an EHS did not reduce the detrimental effects of REST, when this regimen is used for 24 h and the variable responses are evaluated several days after this challenge.

## 5. Conclusions

Broiler chickens from 14 to 35 d of age, allocated in holding pens and added with an EHS in the feed had similar carcass yield and coccidian oocyst excretion but increased LAB and *Clostridium perfringens* counts compared to broilers fed diets added with GPA. Whilst negative control broilers had lower carcass yield, higher *C. perfringens*, LAB, and oocyst excretion. Broilers subjected to REST had reduced growth performance and meat quality. These findings suggest that EHS could be used to increase the carcass yield and beneficial LAB in broilers; but cannot be used to mitigate the deleterious effects of feed restriction.

## Figures and Tables

**Figure 1 animals-09-01101-f001:**
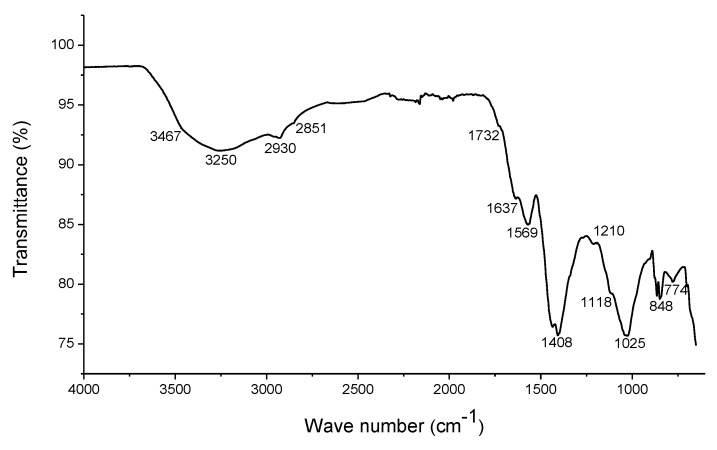
Infrared spectra of the extract of humic substances (EHS). All spectra were collected in a range of 4000–400 cm^−1^, with a resolution of 4 cm^−1^, using 32 scans per sample.

**Table 1 animals-09-01101-t001:** Ingredient composition and nutrient content of the diets.

Item %	From 1–13 d of Age	From 14 to 35 d of Age		
		Positive Control	Negative Control	Extract of Humic Substances
Ground corn	50.2	63.8	63.8	63.5
Soybean meal	40.9	29.7	29.7	29.7
Vegetable oil	4.22	2.33	2.36	2.41
Calcium orthophosphate	1.70	1.23	1.23	1.23
Calcium carbonate	1.49	1.45	1.45	1.45
Vitamins and minerals ^1^	0.70	0.70	0.70	0.70
Sodium bicarbonate	0.25	0.20	0.20	0.20
Salt	0.28	0.45	0.45	0.45
DL-Methionine	0.15	0.10	0.10	0.10
L-Lysine·HCl ^2^	0.12	0.09	0.09	0.09
L-Threonine	0.02	0.00	0.00	0.00
Antibiotic	0.05	0.05	0.00	0.00
Coccidiostat	0.05	0.05	0.00	0.00
Extract of humic substances	0.00	0.00	0.00	0.25
Calculated nutrient content				
Metabolizable energy, kcal/kg	3000	3100	3100	3100
Digestible Lys %	1.19	1.00	1.00	1.00
Digestible Met %	0.46	0.38	0.38	0.38
Digestible Thr %	0.79	0.65	0.65	0.65
Ca %	1.00	0.90	0.90	0.90
Available P %	0.50	0.45	0.45	0.45

^1^ Each kg provided: 6500 IU Vit A; 2000 IU Vit D3; 15 IU Vit E; 1.5 mg Vit K; 1.5 mg thiamine; 5 mg riboflavin; 35 mg niacin; 3.5 mg pyridoxine; 10 mg pantothenic acid; 1500 mg choline; 0.6 mg folic acid; 0.15 mg biotin; 0.15 mg Vit B12; 100.0 mg Mn; 100 mg Zn; 50 mg Fe; 10 mg Cu; 1.0 mg I. ^2^ Synthetic lysine hydrochloride.

**Table 2 animals-09-01101-t002:** Elemental composition of an extract of humic substances ^1^.

Element	Atomic Content %	Standard Deviation	Standard Error
O	45.77	0.59	0.34
Na	27.45	1.386	0.80
C	13.79	0.982	0.567
Si	4.39	0.538	0.311
K	3.88	1.601	0.924
Cl	2.88	0.489	0.282
S	1.50	0.143	0.083
P	0.81	0.059	0.034
Ca	0.43	0.119	0.069

^1^ The sample was measured in triplicate, using zone mapping.

**Table 3 animals-09-01101-t003:** Effect of dietary treatment and feeding regimen on the growth performance and carcass yield of broilers chickens.^a^

Feeding Regimen (FR)	Ad Libitum			Restricted Feeding			SEM ^c^	*p*-Value		
Dietary Treatment (DT)	PC ^b^	NG	EHS	PC	NG	EHS		FR	DT	DT * FR
Initial weight g	342.2	326.2	336.9	337.2	341.8	331.7	14.284	0.88	0.91	0.69
Final weight g	1967.9	1885.5	1909.6	1817.6	1806.3	1791.0	52.465	0.01	0.62	0.79
Feed intake g/d	76.6	73.5	74.2	70.1	69.1	68.8	1.234	0.01	0.81	0.37
Weight gain g/d	98.7	97.6	97.2	92.0	94.2	93.1	2.054	0.01	0.55	0.87
Feed conversion ratio	1.32	1.34	1.33	1.34	1.38	1.37	0.034	0.20	0.66	0.94
Legs %	8.3	8.3	8.3	8.4	8.4	8.5	0.179	0.53	0.82	0.84
Thighs %	9.0	8.8	8.9	8.9	8.8	9.0	0.137	0.72	0.53	0.84
Breast %	24.2	23.6	23.8	23.0	22.8	23.3	0.266	0.01	0.17	0.47
Carcass %	41.5	40.7	41.0	40.4	40.0	40.8	0.314	0.01	0.05	0.31

^a^ Data are means of 30 replications per treatment with one bird/replicate.^b^ PC = positive control; NC = negative control; EHS = Extract of humic substances obtained from a worm compost using an alkaline solution. ^c^ Standard error of the mean.

**Table 4 animals-09-01101-t004:** Effect of dietary treatment and feeding regimen on the tibia measurements and balance of dietary components of broilers chickens. **^a^**

Feeding Regimen (FR)	Ad Libitum			Restricted Feeding				*p*-Value		
Dietary Treatment (DT)	PC ^b^	NG	EHS	PC	NG	EHS	SEM ^c^	FR	DT	DT * FR
**Tibia Measurements**										
Fresh weight g	12.1	13.3	13.0	11.7	12.3	12.1	0.444	0.02	0.14	0.84
Dried matter %	40.7	39.8	42.1	39.0	39.9	39.6	0.871	0.04	0.45	0.26
Dried weight g	4.9	5.3	5.4	4.6	4.9	4.8	0.201	0.01	0.19	0.77
Ashes %	62.8	62.3	64.3	63.3	64.6	62.8	0.918	0.57	0.9	0.13
Ashes weight g	3.1	3.1	3.5	2.9	3.1	3.0	0.153	0.03	0.29	0.18
**Balance of Dietary Components**										
Dry matter										
Intake g/d	127	132	129	130	134	132	2.855	0.25	0.37	0.97
Excretion g/d	36.7	37.9	37.7	39.3	39.3	39.1	0.854	0.02	0.75	0.73
Retention %	70.8	71.1	70.7	69.8	70.5	70.5	0.719	0.3	0.76	0.87
Ashes										
Intake, g/d	13.6	14.0	13.8	13.9	14.2	14.1	0.288	0.25	0.38	0.97
Excretion, g/d	11.3	12.2	11.5	11.5	12.2	11.6	0.327	0.17	0.88	0.56
Retention, %	15.5	13.3	16.7	17.3	14.1	17.7	0.365	0.42	0.46	0.39
Nitrogen										
Intake g/d	3.6	3.7	3.7	3.7	3.8	3.8	0.081	0.25	0.37	0.97
Excretion g/d	1.3	1.3	1.4	1.4	1.4	1.4	0.037	0.01	0.24	0.45
Retention %	64.5	65.4	63.1	63.0	62.2	62.7	0.947	0.03	0.51	0.30
Energy										
Intake Kcal/d	524	543	532	536	550	545	11.746	0.25	0.37	0.97
Excretion Kcal/d	133.9	142.9	138.3	146.2	146.2	146.1	3.566	0.01	0.43	0.43
Retention %	74.1	73.6	73.9	72.7	73.4	73.2	0.743	0.18	0.98	0.67
AMEn Kcal/kg of feed	2903	2879	2899	2847	2880	2869	29.067	0.22	0.96	0.60

^a^ Data of tibia measurements are means of 30 replications per treatment with one bird/replicate and data of balance of dietary components are means of 15 replications per treatment with three pooled samples /replicate. ^b^ PC = positive control; NC = negative control; EHS = Extract of humic substances obtained from a worm compost using an alkaline solution. ^c^ Standard error of the mean.

**Table 5 animals-09-01101-t005:** Effect of dietary treatment and feeding regimen on the meat quality and antioxidant status of breast meat and the counts of lactic acid bacteria, *Clostridium perfringens,* and coccidian oocysts of broilers chickens. **^a^**

Feeding Regimen (FR)	Ad Libitum			Restricted Feeding			SEM ^c^	*p*-Value		
Dietary Treatment (DT)	PC ^b^	NG	EHS	PC	NG	EHS		FR	DT	DT * FR
pH	6.3	6.3	6.3	6.3	6.1	6.2	0.077	0.05	0.25	0.42
Exterior breast meat color										
L* (Brightness)	74.4	74.1	73.8	74.3	75.7	75.4	0.480	0.02	0.47	0.14
a* (redness)	4.9	4.9	5.1	4.7	5.1	5.3	0.259	0.76	0.39	0.71
b* (yellowness)	10.2	9.7	9.7	9.9	10.3	9.9	0.605	0.73	0.92	0.71
Water holding capacity %										
Filtration	15.3	14.5	15.7	14.4	14.0	18.7	1.848	0.51	0.26	0.71
Centrifugation	32.8	37.2	34.2	29.6	23.2	29.3	4.677	0.05	0.94	0.97
Dripping	1.3	0.9	0.9	1.0	1.0	1.0	0.159	0.93	0.26	0.48
Antioxidant status ^d^										
TBARS, mg MDA/kg meat	0.14	0.12	0.14	0.16	0.16	0.14	0.023	0.22	0.97	0.74
DPPH, mmol Trolox/kg meat	16.8	19.1	17.5	16.6	17.4	15.8	2.516	0.18	0.26	0.71
FRAP, mmol Trolox/kg meat	14.5	15.7	16.0	14.7	14.6	13.5	1.065	0.57	0.98	0.96

^a^ Data of meat quality and antioxidant status are means of 15 replications per treatment and data of microbiological status are means of 10 replications per treatment with three pooled samples/replicate. ^b^ PC = positive control; NC = negative control; EHS = Extract of humic substances obtained from a worm compost using an alkaline solution ^c^ Standard error of the mean. ^d^ TBARS = thiobarbituric acid-reactive substances; DPPH = 1,1-diphenyl-2-picrylhydrazyl radical scavenging activity; FRAP = ferric radical antioxidant power.

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
