# Peer review of "Effect of the Addition of Humic Substances as Growth Promoter in Broiler Chickens Under Two Feeding Regimens"

_animals, 2019, doi:10.3390/ani9121101_

Round 1
Reviewer 1 Report
I recommend accepting the manuscript after correctly following Animals guidelines in paper style:
1- Results and discussion should be separated into two parts.
2- References style needs some concern like: the journal abbreviation should be Italic, year should be Bold, ..etc.
Reviewer 2 Report
The manuscript has been improved according to previous suggestions. However, further improvements are required before considering acceptable for publications.
Major concerns
1-Structure of the results and discussion section. I would suggest dividing it into 3 different subsections.
Composition of humic substances. Effect of dietary treatments. Effect of feeding regime.
This partitioning should be preceeded by a statement indicating that a significant interaction between FR and DT was not found in any of the measured variables.
2- The introduction of the manuscript justifies a possible interaction between the FR and DT. Authors have not found it. I miss an explanation in the discussion.
3- The longest section of the results and discussion section is used to describe the DT effect. However, in tables readers find the FR within the DT. My belief is that it would be much more useful for the readers if in tables appeared the DT within the FR.
4- The effect of FR is not described in either the simple summary, the abstract or the conclusion. It should be included.
Other concerns
Line 39-41. Unless a clear link between productive performance and these pararmeters is established without affecting FCR or nutrient retention you cannot state so.
Line 74 and 76. Define these abbreviations. I think it is the first time they appear.
Line 99 and 153. Specify conditions: how long? what stove?
Line 144. How did you slaughter birds? by CO2 inhalation? other?
Line 201- Provide information on the incubator you used
Line 233. Delete quite
Table 1: Delete abbreviation on table heading. Provide this information as supplementary table because it is just informative to understand a result.
Table 2. Delete abbreviation on table heading.
Line 256-258. I agree. Then, why do not you show the effect of DT within FR?
Line 259-261. I think authors want to state that they did not find a significant interaction. If so, please reword, it is not clear.
Table 3. Heading: composition on nutrient content. Reduce decimal use for corn and soybean meal. Define EHS in table caption.
Line 269-271. Please reduce decimals. You found significant differences, but are they relevant from a productive point of view? 40.3 vs 40.9?
Table 5. Reduce decimals in intake and energy. Provide 3 decimals for P values (in this table and elsewhere). EMA or EMAn? Define abbreviations: EMAn, DT, FR...(in this table and elsewhere)
Line 287-291. But you measured FCR and balance of dietary components, and you did not find significant differences for dietary treatments, as you mention in Line 301-302. Authors cannot use the results reported in that study when they are contradictory to what authors have actually found.
Line 308-309. Any hypothesis?
Line 338-340 Any hypothesis?
Line 392-394 without an improvement in FCR or retention? How? you need to establish a clear link
Line 402-403 Would this not have resulted in an improved FCR?
Line 429-431. This is a hypothesis. Please either delete or reword.
Reviewer 3 Report
For the bacterial nomenclature, authors sould use only internationally recognozed abbreviations. For example: Clostridium perfringens mus be abbreviated as C. perfringens or Cp.
The spin speed in Value x g, such as 3000 x g
Page 1
Line 22: Growth promoting antibiotics (GPA).
Line 29: This in the conclusion from your investigation?
Line 31: with
Page 2
Line 66: The name of bacteria, must be written in full the first time is mentioned (Escherichia coli) and then it can be written as E. coli.
coli = Escherichia coli or Entamoeba coli ???
All scientific names of microorganisms must be italicized.
Line 79: This information is irrelevant, since it is not measured in the present study
Page 4
Line 194: Man Rogosa Sharpe medium (MRS)
Line 195: Is Mexico City
Page 7
Line 260: p Value???
Line 262: The table with the composition of the diet should be in the methodology and not in the results section
Page 10
Line 325: improve the design of the table 6, since it is not understandable
Line 326: Italics in C. perfringens name
Page 13
Line 428: Is LAB???
Line 429: GAP

Author Response
Please see the attachment

This manuscript is a resubmission of an earlier submission. The following is a list of the peer review reports and author responses from that submission.
Round 1
Reviewer 1 Report
Page 2
Line 58: The humans are part of the complex of humus substances.
Line 75: What do you mean by better intestinal health?
Line 88: The spin speed should be expressed in g
Page 3
Line 109: It is not clear if the chickens were entered in the cages with 14 and 35 days of age or if they entered with 14 days and left the cages when they were 35 days old.
Line 111: The name of the groups is confusing. If the goal is to investigate the EHS, then the positive group is the one receiving the EHS treatment?
Line 113: In these group, did the broilers receive any antibiotic, BMD or salinomycin?
Page 4
Line 176: How the microaerophilic atmosphere was achieved?
Line 180: The egg yolk was bought or prepared in the laboratory, if it was prepared in the laboratory, how was it made or from what brand was it acquired?
Line 192: Units
Page 5
Line 201: The quality of figure is bad, illegible
Page 6
Line 212: Units. The values should be inside parentesis
Line 214: What kind of sources?
Line 235: Which variables?
Line 236: p value
Page 10
Line 329: When you talking about bacterial counts, you should mention the mean
Page 11:
Line 356: The antiparasitic effect cannot be demonstrated, since no oocyte was inoculated and does not have the basal oocyte data.
Line 363 to 371: This information is irrelevant

Reviewer 2 Report
Extensive English revision is required. The simple summary needs to be shorter. Introduction and discussion should be updated with recent studies like:
Humic acid as a feed additive in poultry diets: a review (2019). Iranian Journal of Veterinary Research, 20(3), 167-172.
L 191: Results and discussion should be separated each in a single part. Provide all tables with P value. Table 3: Relative weights (%) of Thigh, ... are enough. Delete weights (g). No need for very old references. Revise references in/out the text and vice versa.
Reviewer 3 Report
The objective of this trial was to evaluate the effect of humic substances fed under two feeding regimes (not restricted vs restricted) on growth performance, nutrient balance, gut microbiology, meat quality and tibia measurements.The introduction is fairly good and the materials and methods are, in general terms, clearly presented.
My main concern is the presentation of the obtained results.
1) Since one of the main reasons for using factorial designs is to examine interactions and my understanding is that this is what authors are looking for: effect of humic substances will be different under restricted feeding compared to not restricted. Then, it is critical that authors actually do so. Thus, a 3×2 factorial should generally be presented in tables as six data columns with either the second factor means listed within the first factor, or vice versa. This is up to authors decision, but the main effect means should not be pooled since it is always possible for readers to calculate main effect means from the individual means within factors, but not possible to calculate the individual means within factors from pooled means. In addition, there should also be columns to show the P values for each main effect and the interaction (missing in all tables).
2) When you have statistically significant interaction effects, you cannot interpret the main effects without considering the interactions. According to Line 234-236 it seems you found a significant interaction (?). If this is so authors need to rewrite the results and discussion section to describe the interaction not the main effect (as it is right now).
Therefore my decision is to reconsider after major revision. I think authors have done a sound work, but presentation of results needs to be improved. I will be happy to continue the review process after this is addressed.
Other minor comments
Introduction
The introduction is easy to read and well structured in general terms. However, I miss a hypothesis of why HS would affect (impairing or improving) meat quality. Take into account that you mention it as one of your objectives.
Line 58. AH or HA? and AF or FA?
Material and methods
Line 91. Provide information on the mill used to grind
Line 108-116. Provide information on how chicks were managed from day 1 to 14. Did you rear them? under what conditions:temperature, lightning program, reared on floor, vaccinated? Did you use only one concentrate from day 14 to 35? in what form: crumbles, pellets...? In my opinion a full description of the concentrate used should be provided at least as supplementary table.
Line 133. Provide information on the standard procedures: AOAC?
Line 121-122. Why 15 chicks only? should not they be 18?
